# Examining the Utility of Visible Near-Infrared and Optical Remote Sensing for the Early Detection of Rapid 'Ōhi'a Death

**Ryan L. Perroy** [1,2,*], **Marc Hughes** [3], **Lisa M. Keith** [4], **Eszter Collier** [2], **Timo Sullivan** [2] and **Gabriel Low** [5]

1    Department of Geography & Environmental Science, University of Hawai'i at Hilo, Hilo, HI 96720, USA
2    Spatial Data Analysis & Visualization Laboratory, University of Hawai'i at Hilo, Hilo, HI 96720, USA;
     eszterc@hawaii.edu (E.C.); tsull@hawaii.edu (T.S.)
3    College of Tropical Agriculture and Human Resources, University of Hawai'i at Manoa, Hilo, HI 96720, USA;
     mhughes7@hawaii.edu
4    Daniel K. Inouye U.S. Pacific Basin Agricultural Research Center, United States Department of Agriculture,
     Agricultural Research Service, Hilo, HI 96720, USA; lisa.keith@usda.gov
5    University of Alaska at Fairbanks, Fairbanks, AK 99775, USA; low.gabriel.g@gmail.com
*    Correspondence: rperroy@hawaii.edu; Tel.: +1-808-932-7259

**Abstract:** The early detection of plant pathogens at the landscape scale holds great promise for better managing forest ecosystem threats. In Hawai'i, two recently described fungal species are responsible for increasingly widespread mortality in 'ōhi'a *Metrosideros polymorpha*, a foundational tree species in Hawaiian native forests. In this study, we share work from repeat laboratory and field measurements to determine if visible near-infrared and optical remote sensing can detect pre-symptomatic trees infected with these pathogens. After generating a dense time series of laboratory spectral reflectance data and red green blue (RGB) images for inoculated 'ōhi'a seedlings, seedlings subjected to extreme drought, and control plants, we found few obvious spectral indicators that could be used for reliable pre-symptomatic detection in the inoculated seedlings, which quickly experienced complete and total wilting following stress onset. In the field, we found similar results when we collected repeat multispectral and RGB imagery over inoculated mature trees (sudden onset of symptoms with little advance warning). We found selected vegetation indices to be reliable indicators for detecting non-specific stress in 'ōhi'a trees, but never providing more than five days prior warning relative to visual detection in the laboratory trials. Finally, we generated a sequence of linear support vector machine classification models from the laboratory data at time steps ranging from pre-treatment to late-stage stress. Overall classification accuracies increased with stress stage maturity, but poor model performance prior to stress onset and the sudden onset of symptoms in infected trees suggest that early detection of rapid 'ōhi'a death over timescales helpful for land managers remains a challenge.

**Keywords:** Hawai'i; vegetation indices; Ceratocystis lukuohia; Ceratocystis huliohia

## 1. Introduction

Introduced pathogens pose a major threat to forests worldwide, resulting in significant ecological, economic, and cultural costs [1–3]. In the Hawaiian Islands, introduced fungal pathogens are causing a widespread and growing outbreak of mortality in *Metrosideros polymorpha* Gaud. ('ōhi'a), a keystone species in Hawaiian native forests [4–6]. Trees affected by the colloquially termed "rapid 'ōhi'a death" (ROD) display a sudden onset of visible canopy wilting and browning lasting from a period of weeks to months, related to the disruption of their internal vascular system by either of two introduced fungal

pathogens in the genus *Ceratocystis* [7,8]. *C. lukuohia*, the more aggressive pathogen, causes a systemic vascular wilt disease that generally affects the entire tree, while *C. huliohia* is not considered a systemic pathogen and may initially affect only individual branches until enough of the vascular system is compromised to result in canopy death [9,10]. Trees are identified as potential ROD suspects largely by their highly conspicuous visible symptoms (Figure 1), but confirmation of *Ceratocystis* infection requires collecting and analyzing a physical sample for the presence of the pathogens' DNA [11]. Once a tree becomes symptomatic, the disease is terminal, and the infected tree may become a source for further transmission [12].

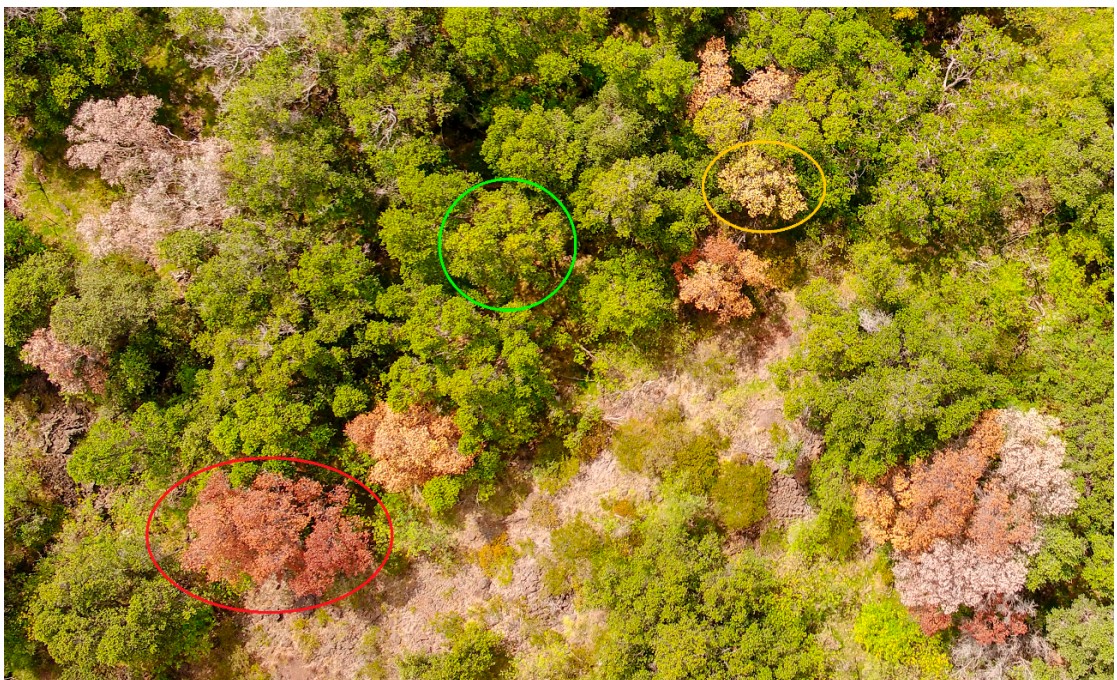

**Figure 1.** Small unmanned aerial system (sUAS) image (October 9, 2017, taken at 12:58 Hawaii Standard Time (HST) at 120 m altitude) showing "rapid 'ōhi'a death" (ROD)-affected 'ōhi'a trees on Hawai'i Island (19°22'13" N, 155°22'20" W) in various stages of expression of visible symptoms. Colored circles depict example asymptomatic (green), early onset (yellow), and late stage (red) 'ōhi'a trees.

Early detection is of great interest across the entire Hawaiian archipelago, where native forests are already under the combined pressures of aggressive introduced species and climate change [13,14], but particularly in areas where ROD has yet to establish itself [15]. On a landscape basis, symptomatic trees have been successfully mapped via visible and hyperspectral remote sensing [6,15,16], but there is no present means of detecting asymptomatic trees infected with the fungal pathogens responsible for ROD or of definitively discriminating ROD-affected trees from trees experiencing other stressors, including drought. Elsewhere, hyperspectral remote sensing has been successfully used for the early detection of other forest fungal pathogens, including oak wilt [17], myrtle rust [18], and laurel wilt [19–21]. Thermal remote sensing, either alone or in combination with hyperspectral data, has been used to detect a variety of forest stressors [22–24]. Other methods of early detection for forest pathogens, including canine olfaction [25] and gas chromatography/electronic noses [26], may also hold promise for ROD.

Here, we present work from laboratory and field inoculation trials, examining the potential for using hyperspectral and red green blue (RGB) visible wavelength imagery at the leaf and branch scale to detect ROD prior to the onset of visible symptoms. Our goals for this study were to:

1.  Characterize spectral progression changes for the two ROD fungal pathogens, *C. lukuohia* and *C. huliohia,* and extreme drought conditions, through frequent spectroradiometer measurements and RGB photography;
2.  Determine if the collected datasets, including derived simple vegetation indices (VIs), can be used to detect and discriminate between treated but visually asymptomatic ʻōhiʻa trees; and,
3.  Examine how to effectively translate laboratory results into practical field-deployable diagnostic tools for the early detection of ROD across the Hawaiian Islands.

## 2. Materials and Methods

### 2.1. Laboratory Measurements

Approximately two- to three-year-old *M. polymorpha* seedlings were randomly placed in a growth chamber (Controlled Environments Ltd., Winnipeg, Canada) set to 23 °C and 85% relative humidity with a 16-h photoperiod (Figure 2A). Seedlings in the inoculation trials were watered to field capacity every other day. Twelve seedlings were inoculated twice on the main stem with fungus-colonized filter paper discs (six with *C. huliohia* and six with *C. lukuohia*), following the methods outlined in [8] (Figure 2B). Six seedlings inoculated with a filter paper disk soaked in sterile distilled water were used as negative controls.

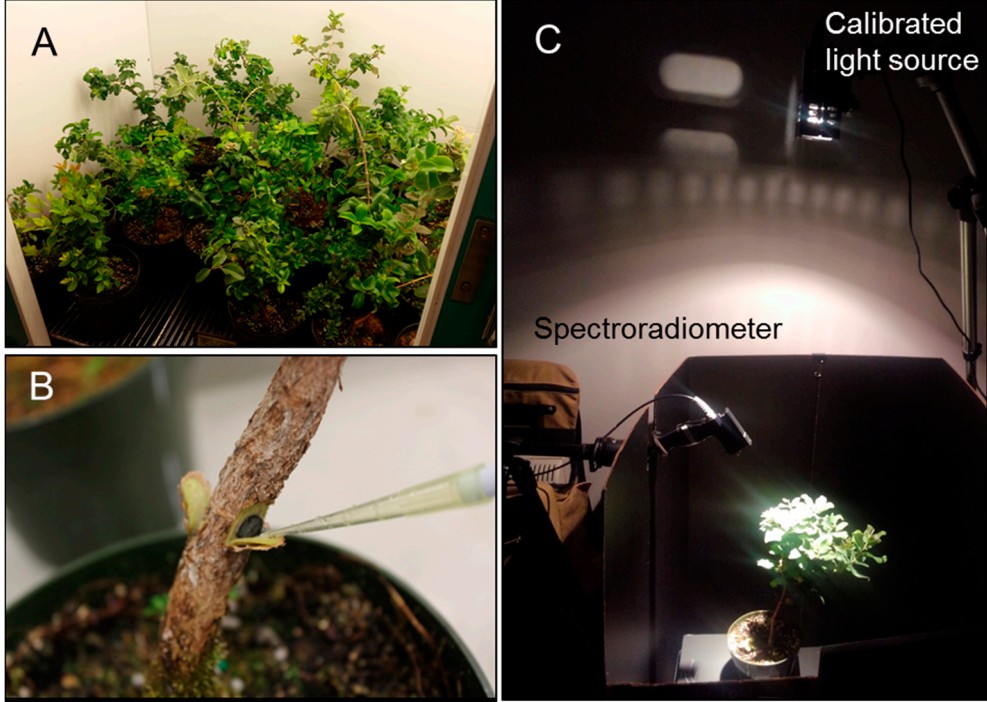

**Figure 2.** Laboratory experimental set-up. (**A**) ʻōhiʻa seedlings in the growth chamber, (**B**) inoculation procedure for an ʻōhiʻa seedling, (**C**) spectral data collection measurement chamber.

Data collection started 16 days prior to inoculation and continued throughout the trial at 2–4-day intervals. Plant health and vigor was assessed visually every two days using a foliar wilt rating scale: 0 = no wilt, 1 = 0–20%, 2 = 21–40%, 3 = 41–60%, 4 = 61–80%, 5 = 81–100%. The first visual observation of wilt (first date a plant received a score of 1) was used to establish laboratory stress onset, even if the wilting occurred in peripheral leaves outside the imaging area of spectral data collection.

Plants were photographed using a Canon Powershot SX610 HS 20.2 MP camera from two tripod mounted fixed-distance look directions (top-down from above the measurement table and from the side). A 24-color CameraTrax calibration card was included in the frame for each photograph. A white grease pen was used to mark plant containers and their corresponding placement location on the

measurement table to ensure a consistent position and orientation for the photographs and spectral measurements as leaf inclination and plant geometry illumination effects can have significant impacts on spectral measurements [27,28]. All laboratory measurements were conducted at the United States Department of Agriculture Agricultural Research Service (USDA ARS) facility in Hilo, HI, USA.

Spectral measurements were collected using a bare-fiber FieldSpec Pro spectroradiometer and 70 W 3100 K Quartz–Tungsten–Halogen light source for indoor diffuse reflectance measurements (ASD Inc., Boulder, CO, USA). The spectroradiometer and light source were turned on 1–2 h prior to data collection to warm up and ensure stable operating conditions. Spectral measurements were made in a darkened room and the plants placed within a measurement chamber lined with flat black cardboard. The spectroradiometer bare fiber (25° field of view) was placed in a pistol grip holder at a 45° relative to the top of the plants and mounted to a tripod; the light source was positioned directly above the plants (Figure 2C). As the plants were different heights, platforms were used to raise and lower the plants to maintain a distance of 10 cm from the pistol grip holder to the top of the plant surface, resulting in a 4.43 cm diameter circular collection area that was consistently imaged across the duration of the experiment. A leaf clip was initially used to collect leaf-scale spectral measurements, but the clip and light source damaged the leaves so this effort was abandoned for the non-contact spectral measurements described above.

Prior to each plant spectral measurement, a 12.7 cm × 12.7 cm white Spectralon reference panel (Labsphere, North Sutton, NH, USA) was placed in the measurement chamber at the same distance (10 cm from the instrument) and measured to establish the 100% reflectance baseline condition. The reference panel was then quickly removed, and the plant placed in its proper orientation and position. Spectral measurements were then carried out (25 measurements per plant), and the white reference re-measured every 5 min.

After 89 days, the inoculation experiment was stopped as all plants had fully wilted and a second trial commenced to measure the spectral impacts of drought on 'ōhi'a seedlings. This was done to examine differences between extreme drought-affected plants and plants inoculated with the two fungal pathogens responsible for ROD, as these conditions may similarly affect internal water transport [29]. Limited seedling availability did not allow us to run the inoculation and drought trials concurrently. Twelve 'ōhi'a seedlings were chosen for extreme drought (complete cessation of watering) and six seedlings were watered normally as a control treatment. Plant health and vigor were assessed visually every two days by a trained observer using the same wilt rating scale described earlier. Spectral and photographic data collection started 18 days prior to the cessation of watering for the drought-affected plants and continued throughout the trial at a 2–4 day interval. We recognize that complete cessation of water is an extreme form of drought, and that 'ōhi'a possess drought tolerance traits (selective leaf senescence, e.g.) that may cause them to respond differently to partial drought conditions [30].

*2.2. Data Processing*

Spectral time series measurements for all plants were exported into text-readable files using ViewSpec Pro (Malvern Panalytical, Malvern, UK) and then all processing and analyses were done in MATLAB (ver. 2019b, Natick, MA, USA). Raw spectra were processed for jump correction [31] and brightness normalization [32,33] to minimize variability unrelated to plant stress.

Four well-established spectral vegetation indices (VIs) were generated from these data to examine their sensitivity for detecting ROD and drought stress in visually asymptomatic 'ōhi'a seedlings (Table 1). These VIs, the Cellulose Absorption Index (CAI), Moisture Stress Index (MSI), Normalized Difference Vegetation Index (NDVI), and Photochemical Reflectance Index (PRI), are sensitive to plant senescence, disease, and/or water stress and involve wavelength regions previously identified as important for discriminating between ROD-affected and healthy trees [16].

**Table 1.** Vegetation Indices (VIs) included in the study.

| Data Source | Index | Formula | Reference |
|---|---|---|---|
| Spectroradiometer | NDVI | $(R_{800} - R_{670})/(R_{800} + R_{670})$ | [34,35] |
| | PRI | $(R_{570} - R_{531})/(R_{570} + R_{531})$ | [36] |
| | MSI | $R_{1600}/R_{820}$ | [37] |
| | CAI | $0.5 * (R_{2015} + R_{2195}) - R_{2106}$ | [38] |
| RGB Camera | ExG-ExR | $(2 * g - r - b) - (1.4 * r - g)$ | [39] |
| | VCI | $g/(r + b)$ | [40] |

The CAI, which measures a cellulose absorption feature near 2100 nm, was originally developed to categorize crop residue cover and is strongly attenuated by the presence of water [38]. The MSI simple reflectance ratio [37] has been used to detect moisture stress in various settings [41,42], including pre-visual water stress detection in potato plants [43]. The PRI, a widely used carotenoid index [36], was developed to capture the shift from violaxanthin to zeaxanthin as stressed plants are no longer able to use light absorbed by chlorophyll and is a good surrogate for light use efficiency in stressed plants [44]. The NDVI [34,35] is a well-established and widely used VI. If effective for the early detection of ROD, these types of simple VIs could be feasibly included as part of a large-scale ROD monitoring program.

Raw RGB photos were individually processed to extract a user-defined region of interest ($100 \times 100$ pixels minimum) from a representative leaf-cluster within the spectroradiometer measurement area. Extracted RGB pixel values were used to compute two different VIs: excess green minus excess red (ExG-ExR) [39] and the vegetation contrast index (VCI) [40]. Recent work has shown VIs derived from consumer-grade RGB cameras were comparable or better than spectroradiometer-derived VIs in estimating chlorophyll content and other parameters for wheat [45]. The ExG-ExR index and its constituent components, excess green (ExG) [46] and excess red (ExR) [47], were originally developed to isolate vegetation from non-vegetation in RGB photography, but the index has also been used successfully to detect vegetation stress from RGB imagery collected from a small unmanned aerial system (sUAS) [48]. The simple VCI index has been shown to be sensitive to detecting phenology changes, including the start of senescence [40].

*2.3. VI Stress Onset Detection*

We compared each VI's ability to detect stress onset (SO) for the different experimental treatments, to determine their utility for early detection. The date of SO was determined two different ways for each complete VI curve (Figure 3). In the first approach, SO was determined by finding the inflection point along the VI stress curve indicating departure from a healthy condition, similar to methods employed to detect phenological transitions from satellite remote sensing data [40]. VI time series data were smoothed using Savitzky–Golay filtering [49] with a second order polynomial and a frame length of five, and then fit with a shape-preserving Piecewise Cubic Hermite Interpolating Polynomial (PCHIP) function [50,51] to generate interpolated values at a 1-day interval. The maximum first derivative value of these data (period of greatest spectral change) was used to identify the "center" of the active stress period, and a local search then used to identify the SO as an abrupt curvature change point prior to the stress center [52,53]. In the second approach, a simple 20% threshold in the stress direction (positive for MSI, negative for all other indices) was used for each plant relative to a pre-treatment baseline VI condition. The 20% threshold value was chosen to minimize false triggers from noise in the datasets.

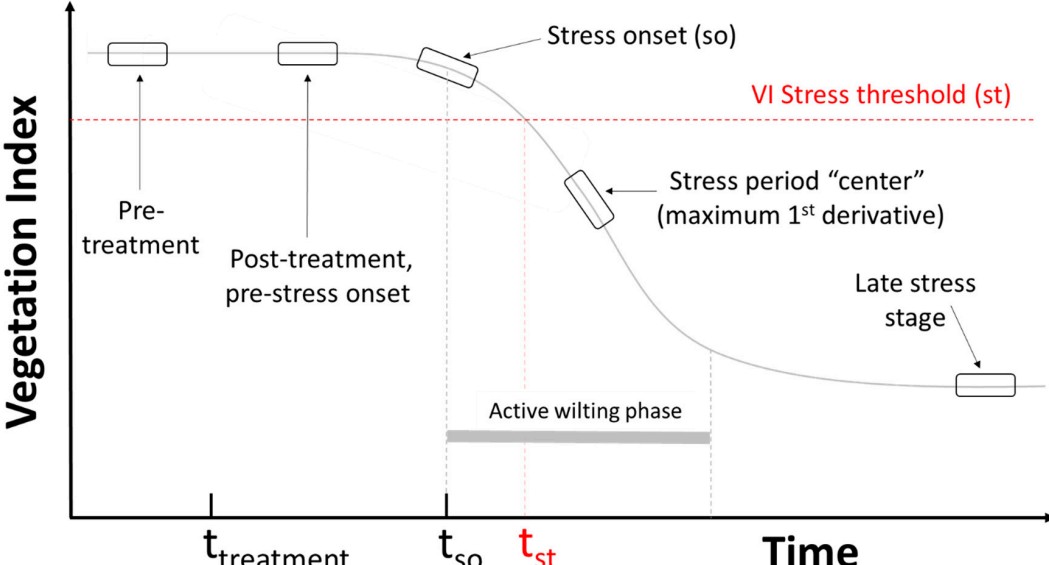

**Figure 3.** Idealized vegetation stress curve and important phases and transitions. Curve is for a VI that decreases in value with increasing stress, such as Normalized Difference Vegetation Index (NDVI). $t_{treatment}$ is the time of treatment, $t_{so}$ is stress onset (based on first inflection of VI curve), and $t_{st}$ is the time of crossing the VI stress threshold value. The horizontal bar signifies timing and duration of active wilting phase, beginning with stress onset (SO) inflection point. The vertical position of the horizontal bar is arbitrary.

To examine the utility of using incomplete time series datasets for SO detection, we also searched for the first appearance of an abrupt curvature change point in the VI curve time series, beginning with the first three time series data points and an incrementing for loop to sequentially increase the amount of data available for analysis, resulting in progressively longer curves to search.

## 2.4. Field Measurements

To help determine how the laboratory seedling results translated to a forest setting with mature trees, we took advantage of a kiln-heating inoculation study in the Waiakea Forest Reserve on Hawai'i Island, an area already heavily impacted by both *C. huliohia* and *C. lukuohia* (Figure 4). Fourteen trees from that inoculation study were monitored over time (three with *C. lukuohia* and eleven with *C. huliohia*), along with ten nearby non-inoculated trees for controls. Tree diameter at breast height (dbh) ranged from 20–33 cm. A small unmanned aerial system (sUAS), a Matrice 200 (DJI Inc., Shenzhen, China) was used to collect repeat visible and multispectral imagery over the inoculated trees. sUAS, carrying a variety of sensor payloads, have been used to detect a growing number of forest pathogens [54–56], and provide an efficient means of generating repeat spatial data over monitoring plots. For high resolution RGB visible imagery collection, the imaging payload was a DJI Zenmuse X5S camera (CMOS, 4/3", 20.8 MP). For multispectral imagery collection, the payload was an Altum 6-band sensor (Micasense, Seattle, WA, USA) with center wavelengths of 475, 560, 668, 717, and 840 nm for the non-thermal bands using a fixed exposure of 0.1 ms and gain of 1. Flights occurred between 1000–1400 HST from 20 August 2019 to 8 January 2020, with an average return interval of 9 days (min = 5 days, max = 14 days). Flights were conducted in accordance with the Federal Aviation Administration regulations under Part 107 rules at an altitude of 120 m, resulting in a ground sampling distance of 2.7 cm for the visible imagery and 5.2 cm for the multispectral imagery.

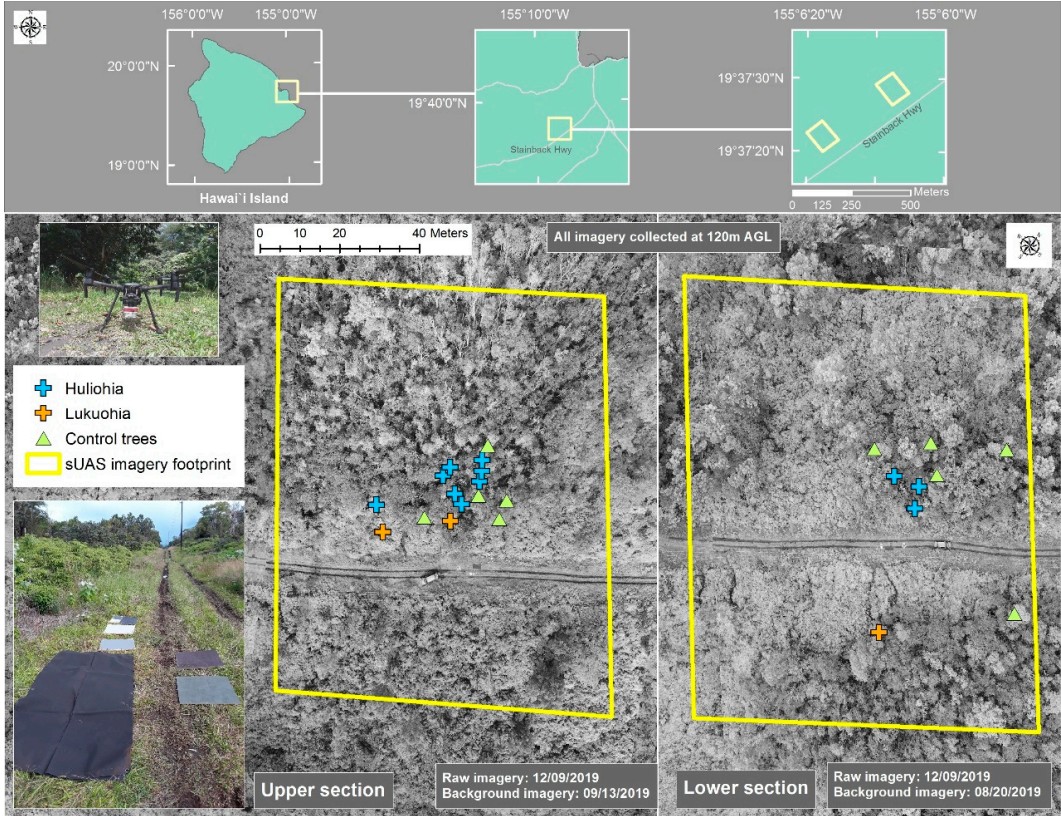

**Figure 4.** Study site, tree locations, upper and lower areas, footprint of single Altum/red green blue (RGB) image.

A series of calibration targets were deployed prior to each multispectral flight to aid in the conversion of raw digital number values into reflectance measurements. Reflectance targets were made out of coated Type 822 woven polyester substrate (Group 8 Technology, Provo, UT, USA) secured over level plywood bases placed within the imaged area. Reflectance properties of each calibration target were independently characterized by spectral measurement via the same spectroradiometer and white reference panel used in the laboratory experiment and those values then used to correct the raw multispectral imagery using empirical line calibration [57,58]. As the study area was small, and East Hawai'i Island atmospheric conditions are highly unstable and variable over short time periods, we used and calibrated a single image for each data point in the time series, instead of orthomosaics derived from images taken at different times under varying cloud conditions that include processing artifacts [59,60].

### 2.5. Aerial Image Processing and Analysis

Raw multispectral bands (red, 668 nm and NIR, 840 nm) were processed to generate a time series of NDVI raster images. The Altum multispectral camera generates individual frames for each band, slightly offset due to the configuration of the sensors. To correct this offset, we manually registered each red band image to its corresponding NIR band image, resulting in an average registration error of <1.7 pixels. We used the in-scene calibration targets and the empirical line calibration method [58] to convert the raw pixel values into reflectance using ENVI image processing software (Harris Geospatial, Boulder, CO, USA). Calibrated red and NIR images were then used to create NDVI rasters via band math in Arcmap (ESRI, Redlands, CA, USA) which were then georeferenced to 2.7 cm resolution RGB orthomosaics generated from UAS flights conducted within 1–3 days of each corresponding multispectral flight. For each canopy crown in the study (13 inoculated trees, 10 control trees), a 1 m diameter circular sample zone region of interest (ROI) was defined and used to extract pixel values.

Sample zone ROIs were manually adjusted for each NDVI raster in the time series to ensure consistent crown center sampling of NDVI pixels, which were analyzed to produce mean and standard deviation NDVI values for each tree on each sample date.

Raw RGB images from the DJI Zenmuse X5S camera were processed to generate a time series of ExG-ExR VI values for each tree crown. Two RGB images (20 September 2019 and 15 October 2019) were collected under heavy cloud cover in low-light conditions and were adjusted by simple histogram matching to the nearest image in the time series. No other image calibration was performed. Each RGB image was georeferenced to a corresponding orthomosaic using the methods described previously and ExG-ExR raster images created in ArcMap. ROI sample zones were generated and adjusted using the same methods as for the NDVI rasters and pixel values extracted to generate mean and standard deviation ExG-ExR values.

## 2.6. Field Sample Collection

All trees involved in the kiln treatment inoculation study were felled beginning on 9 January 2020, thus ending the imagery collection effort. At the time of felling, five of the *C. huliohia* inoculated trees (215, 216, 217, 218, and 219) had yet to exhibit any signs of stress. Branch samples with intact leaves from these trees were collected immediately after felling and transported to the USDA ARS research facility in Hilo, HI for spectroradiometer measurements and RGB photo collection, using the same laboratory methods described above.

## 2.7. Treatment Discrimination via Classification

We examined the ability of linear support vector machine (SVM) learning, applied to the laboratory spectroradiometer and RGB photography datasets, to discriminate between the different treatments: control, *C. lukuohia* inoculation, *C. huliohia* inoculation, and extreme drought at vegetation stress stages ranging from pre-treatment to late stage stress (Figure 3). Linear SVM models classify data using multidimensional decision boundaries, optimized on the basis of user-provided input data. SVM and other machine learning classifiers have proven successful in detecting other plant pathogens using hyperspectral data [17,61,62]. Spectra from 440–2400 nm were down-sampled to 10 nm resolution to reduce the number of predictor variables (from n = 1960 to n = 196), coded by treatment, and grouped by vegetation stress stage, using each plant's MSI curve as the basis for stress stage identification. Extracted average pixel values from the corresponding RGB photographs were used to develop independent classification models at these same stress stages. SVM classification trials were performed in MATLAB R2019b (MathWorks Inc., Natick, MA, USA) using fivefold cross-validation partitioning to protect against overfitting. We generated two-, three-, and four-class models for 12 stress stages, ranging from pre-treatment to late-stage stress. These two-class models (ROD vs. control), three-class models (ROD vs. drought vs. control), and four-class models (*C. lukuohia* vs. *C. huliohia* vs. drought vs. control) were built to assess our ability to discriminate between an increasing number of classes at the twelve different time steps relative to MSI stress onset. The "ROD" class in the two- and three-class models combined *C. lukuohia* vs. *C. huliohia* spectra, which were separated only in the four-class models. We were most interested in assessing model performance at SO and pre-SO stages, for the purposes of early detection, and in comparing the results from spectroradiometer measurements and RGB imagery.

## 3. Results

### 3.1. Laboratory Trials

Spectral signature time series data for all seedlings involved in the study were merged into a database and analyzed with custom scripts in MATLAB. Raw time series spectral data from individual representative plants from each of the four treatments (*C. lukuohia, C. huliohia*, drought, and control) are shown below, along with corresponding side-view RGB photographs from pre-treatment, stress onset (SO), and late-stage phases of the disease or drought (Figure 5). A representative time series of

top-down RGB photos (used to generate ExG-ExR and VCI VIs) for one of the inoculation experiment plants (plant # 8, *C. huliohia*) is shown in Figure 6.

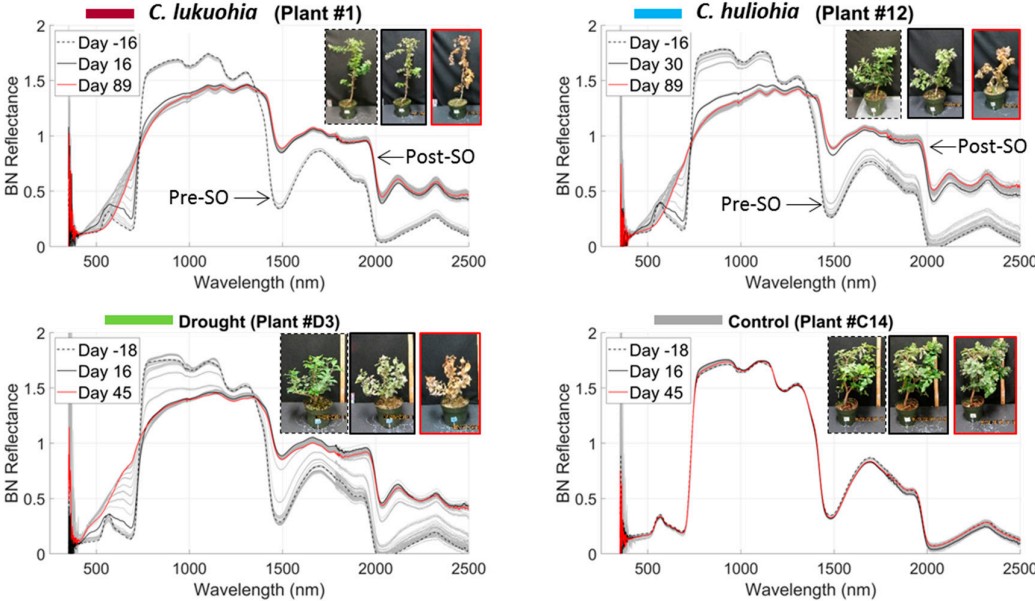

**Figure 5.** Example brightness normalized (BN) spectral time series data for four representative plants from the laboratory trials. Inset images are side-view RGB photos corresponding to pre-treatment (dashed line), stress onset (SO, solid black line), and late stress (red line) stages for treated plants, and corresponding time steps for the control plant. The numbers in the legend refer to days since treatment began, with negative values occurring pre-treatment.

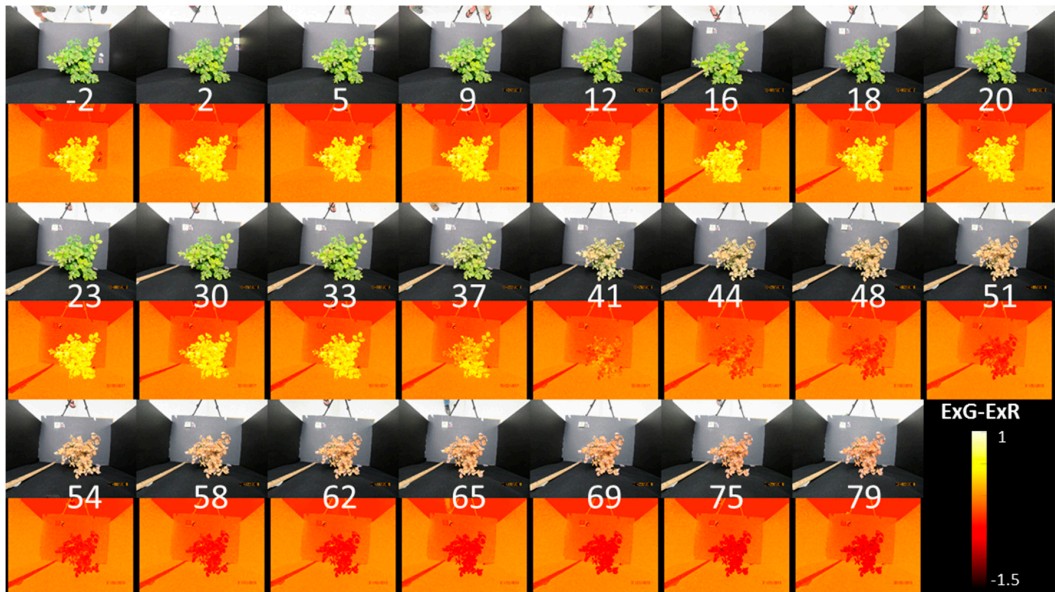

**Figure 6.** Time series of top-down RGB photos and derived excess green minus excess red (ExG-ExR) images for *C. huliohia*-inoculated study plant # 8. Days since inoculation shown in white. Similar images were collected for all other plants in the laboratory experiment.

Time series data for the treated plants reveal a spectral progression from chlorophyll-rich photosynthetic vegetation at field capacity (distinct green peak in the visible, near-vertical red edge leading to higher NIR reflectance values and strong water absorption features) to dry, late-stage non-photosynthetic vegetation (no green peak, red reflectance values gently sloping into the NIR

region with minimal water absorption features). This progression broadly matches vegetation stress sequences observed in other studies [19,63] with some notable differences. At wavelengths >725 nm, the spectral progression for the inoculated seedlings is quite abrupt, revealing two distinct groupings (pre-SO and post-SO) with relatively few to no "transition" spectra before or after. The extreme drought treatment time series has more transition spectra at these wavelengths. At wavelengths <725 nm, all treatments show transition spectra from SO into the late-stress stage, and some transition spectra prior to SO. Post-SO spectral peaks occur at 2125 nm and 2325 nm for all treatments, suggesting that they are not unique ROD-specific features as reported by [16].

Processed spectral datasets were used to generate a number of different VIs (Table 1) for all plants included in the laboratory study (Figure 7). One of the plants in the drought experiment, plant T1, exhibited symptoms indicative of root rot prior to the start of the introduced drought. Given that we could not confidently separate the root rot disease symptoms from the drought symptoms, those results were held back from additional analyses. Most of the VIs clearly separated treated and control plants except for late-stage PRI.

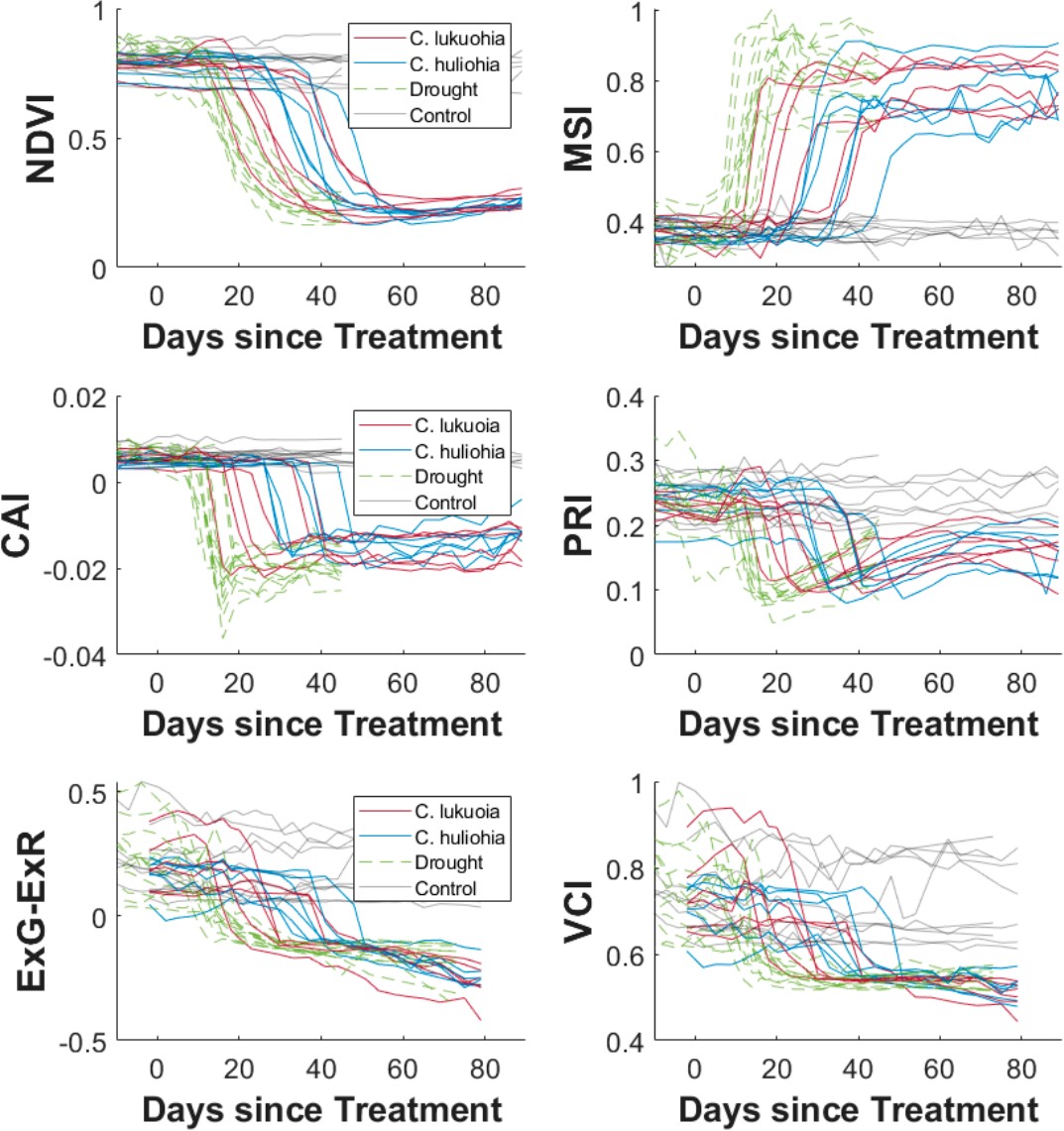

**Figure 7.** Time series of Vegetation Index (VI) curves for all plants involved in the laboratory trials. ExG-ExR and vegetation contrast index (VCI) VIs were derived from RGB photos, all other VIs were derived from spectroradiometer data.

There was significant variation in the timing of stress onset in the inoculated plants, ranging between 10–40 days (mean = 26 days) after inoculation based on a visual determination of first wilting. *C. lukuohia*-inoculated seedlings expressed stress onset an average of 10 days earlier than *C. huliohia*-inoculated seedlings, corresponding to the more virulent nature of that fungal pathogen [7,9]. Drought stress onset occurred more quickly and consistently, ranging between 4–15 days (mean = 8.6 days) after cessation of watering.

To evaluate the performance of individual VIs in detecting stress onset, VI curves were plotted for each plant (Figure 8, Supplementary Figures S1 and S2). The stress onset date and duration of the active wilting phase for each VI, as determined from change detection analysis, are shown as horizontal colored bars in Figure 8. For plant # 7 (*C. huliohia*-inoculated), none of the VIs detected stress onset prior to the visual observation of first wilting at day 40, as indicated by the dashed vertical line.

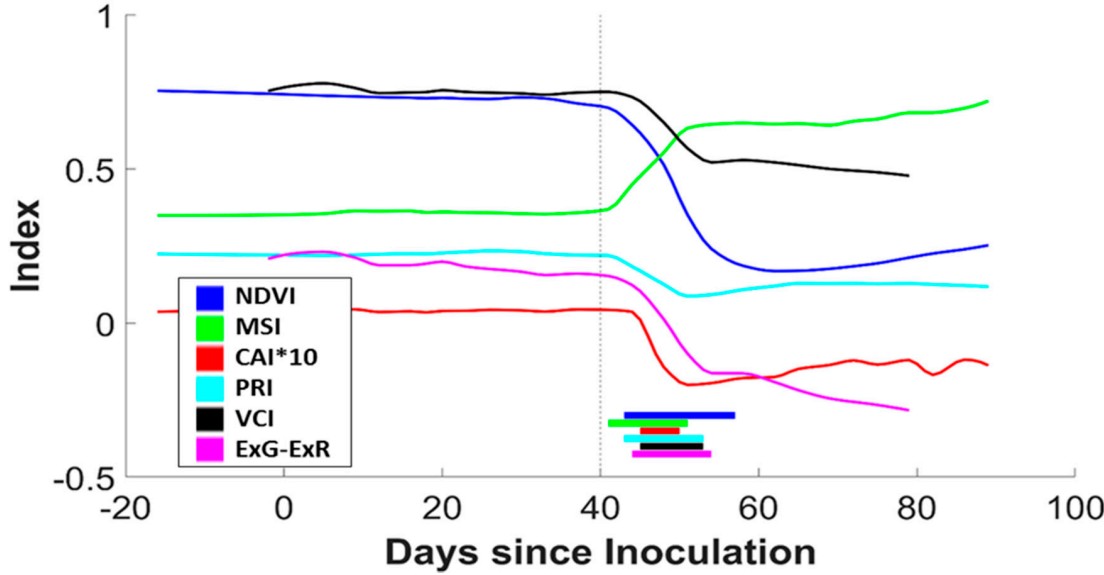

**Figure 8.** Smoothed VI curves for plant #7 (*C. huliohia* inoculation). First laboratory observation of wilting indicated by dashed vertical line. Colored horizontal bars represent onset and duration of active wilting phase, based on change detection analysis of the VI curves as illustrated in Figure 3. VI curves for all plants included in the laboratory trials are shown in Supplementary Figures S1 and S2.

For many of the plants, some or all of the VIs did detect SO prior to the visual observation of first wilting (Figure 9), though never more than five days in advance of visual detection. Of these VIs, Moisture Stress Index (MSI) and Cellulose Absorption Index (CAI) consistently outperformed the others, although there were no significant differences in the average SO detection dates among any of the VIs based on one-way analysis of variance (ANOVA, P = 0.35), when considered relative to the time of first observed wilt (Figure 10). The detection of SO via the curvature change point approach was always more sensitive than the 20% simple threshold for the inoculated seedlings, as indicated by the black dots in Figure 10A. Similar results were seen for the drought treatment (Figure 10B) except for the PRI index, which was triggered early in four out of the twelve seedlings (Supplementary Figure S2), due to signal noise and a relatively small range (Figure 7).

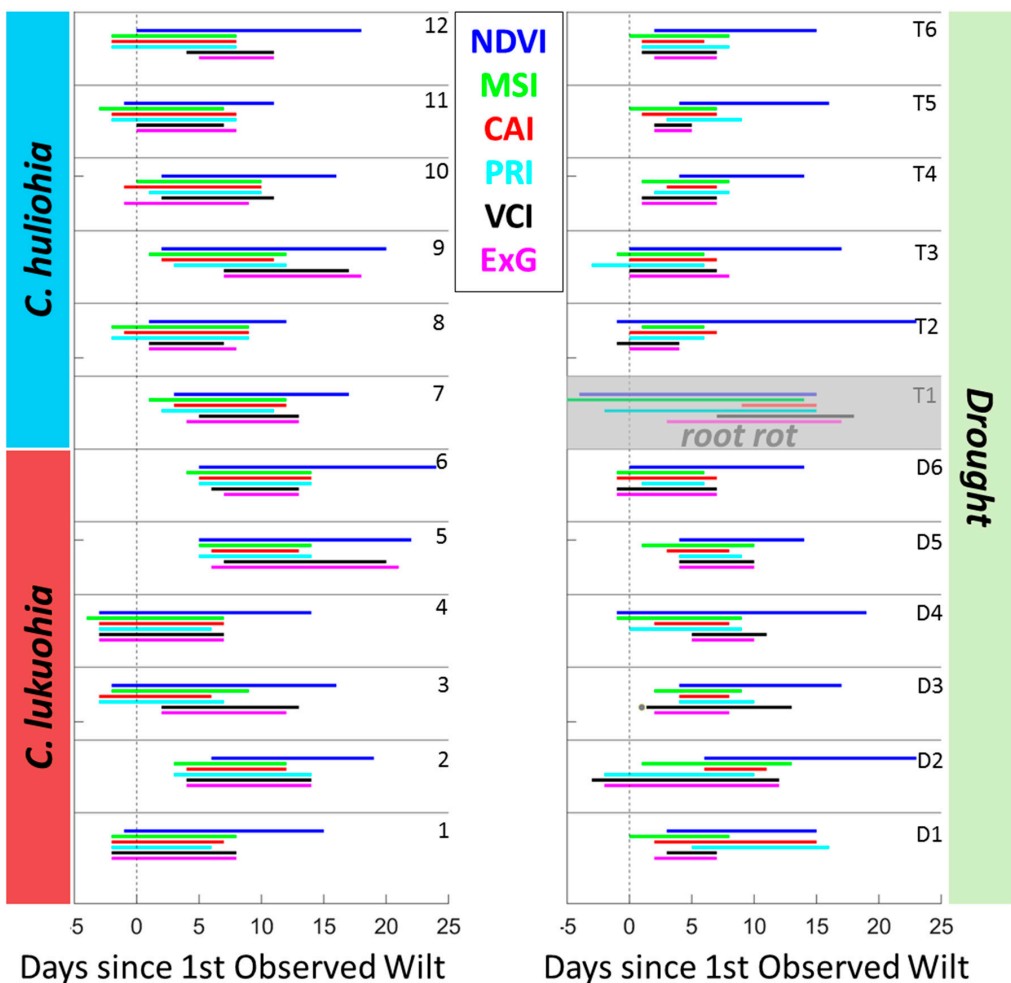

**Figure 9.** Horizontal bars representing onset and duration of active wilting phase for laboratory trials (inoculated treatments on left, drought treatment on right) as determined by different VIs, relative to first laboratory observation of wilting (dashed vertical line). NDVI in blue, Moisture Stress Index (MSI) in green, Cellulose Absorption (CAI) in red, Photochemical Reflectance Index (PRI) in cyan, VCI (from RGB photos) in black, and ExG-ExR ("ExG" from RGB photos) in magenta. Plant T1 exhibited early stress due to unrelated root rot symptoms.

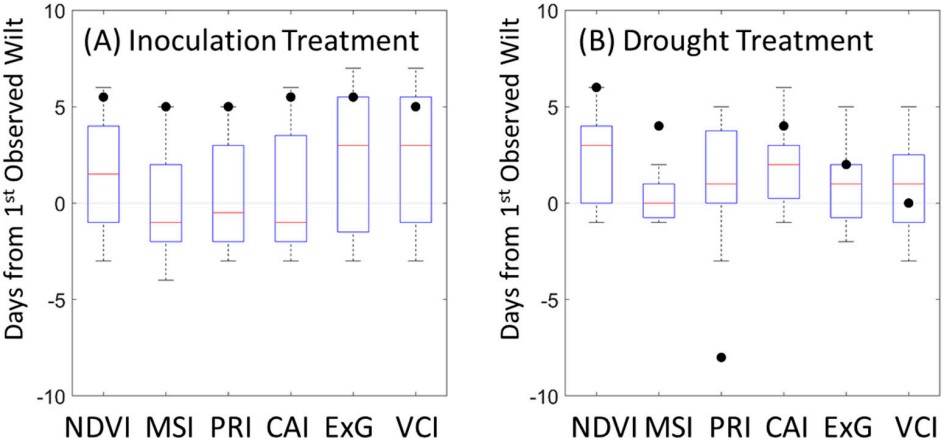

**Figure 10.** Boxplots of stress onset detection dates for the different VIs, relative to first observed wilt (time zero). (**A**) Inoculated treatment on left, (**B**) drought treatment on right. Negative values indicate the detection of stress prior to first recorded observation of wilt. Black dots represent the median date (relative to first observed wilt) each VI curve passed a 20% threshold value.

As both of these SO detection methods (finding an abrupt curvature change point prior to the stress center and applying a simple 20% threshold) depend on VI stress curves that extend into the late-stress stage for analysis, they have limited utility for true early detection. Results from our SO detection analysis using incomplete time series datasets, where we searched for the first appearance of an abrupt curvature change with progressively longer VI curves, can be seen in Supplemental Figure S3. We found that SO detection using this method was quite stable (varying by 1–2 days) in cases with well-behaved curves, and SO typically fell within a few days of what was found using the full VI curve approach. In more noisy datasets, as for seedling # 3 (Figure S3C), we did see early false triggers and more spread in SO dates.

### 3.2. Field Inoculation Trials

We conducted >60 individual sUAS flights between 9 September 2019 and 8 January 2020, over the upper and lower sections of the study area, resulting in a dense time series dataset of visible RGB and multispectral imagery for the 4.5-month period from inoculation to tree felling (Figure 11). Of the fourteen inoculated trees (11 *C. huliohia* and 3 *C. lukuohia*), eight were symptomatic at the time of felling with NDVI and ExG-ExR VI stress curves similar to those in the laboratory inoculation experiments (Figure 12, top panels). Laboratory testing of samples from the felled trees indicated that, of the eleven trees originally inoculated with *C. huliohia*, 64% were naturally co-infected with *C. lukuohia*, the more virulent pathogen. All but one of these co-infected trees, and all of the *C. lukuohia* inoculated trees, exhibited symptoms with the remaining five *C. huliohia* remaining pre-symptomatic at the time of felling.

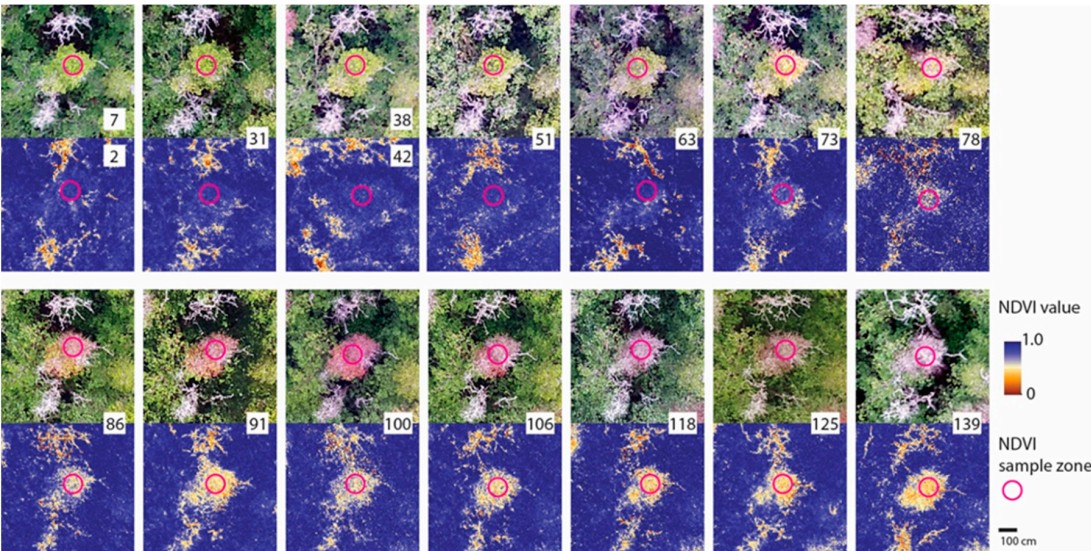

**Figure 11.** Time series of RGB (top) and NDVI (bottom) sUAS paired images for Tree 214 in the field inoculation trial. The numbers indicate the number of days since inoculation.

Of the two VIs derived from the field inoculation trial sUAS imagery, NDVI was able to consistently detect stress onset prior to ExG-ExR, but only by a moderate amount (4.5 days earlier on average, maximum early detection of 18 days).

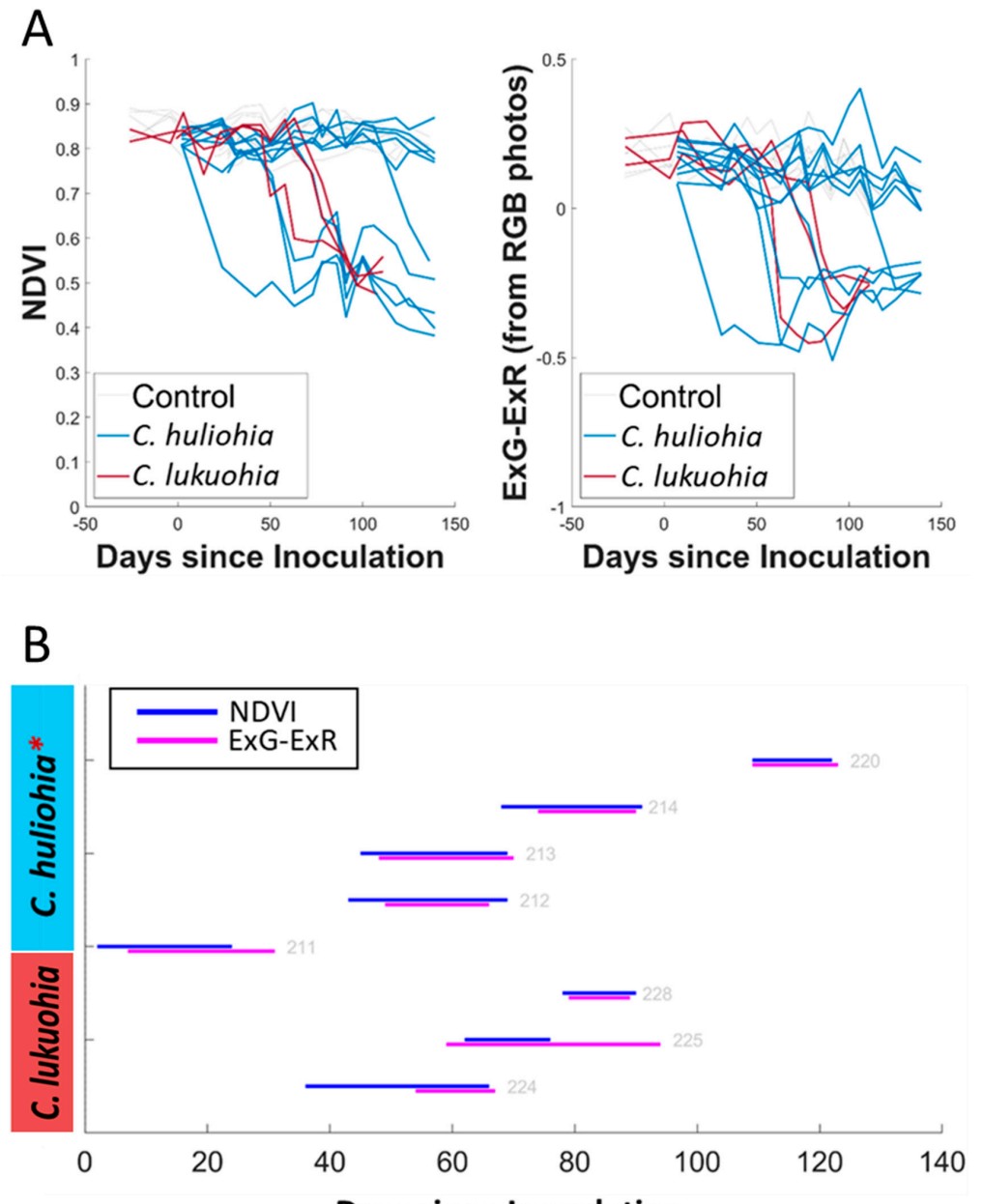

**Figure 12.** (**A**) Time series VI curves for NDVI (top left) and (top right) ExG-ExR from sUAS imagery of all trees monitored in the Waiakea Forest Reserve inoculation field trial. (**B**) Horizontal bars representing onset and duration of active wilting phase from NDVI (blue) and ExG-ExR (magenta) VIs for affected trees. * Symptomatic trees originally inoculated with *C. huliohia* were determined to be co-infected with *C. lukuohia* upon laboratory testing.

### 3.3. Classification Model Results

A sequence of linear support vector machine (SVM) classification models of increasing class size was produced from the laboratory trial 10 nm resampled spectroradiometer data and the RGB image samples. Two-class models (ROD vs. control), three-class models (ROD vs. drought vs. control, and four-class models (*C. lukuohia* vs. *C. huliohia* vs. drought vs. control), were produced for the 10 nm data and RGB imagery at twelve different time steps, ranging from pre-treatment to late-stage stress (Figure 13, Supplemental Table S3). The models generated at each time step were independent from one another and used input data based upon each seedling's appropriate MSI curve position. In this scheme, the SO model (Day 0) was based on data collected at the time of SO, as defined by the change

in the detection analysis of each seedling's MSI curve (Figure 8). Pre-SO models were generated from data collected at time steps prior to the MSI curve SO position, and post-SO models were generated from data collected after the MSI curve SO position.

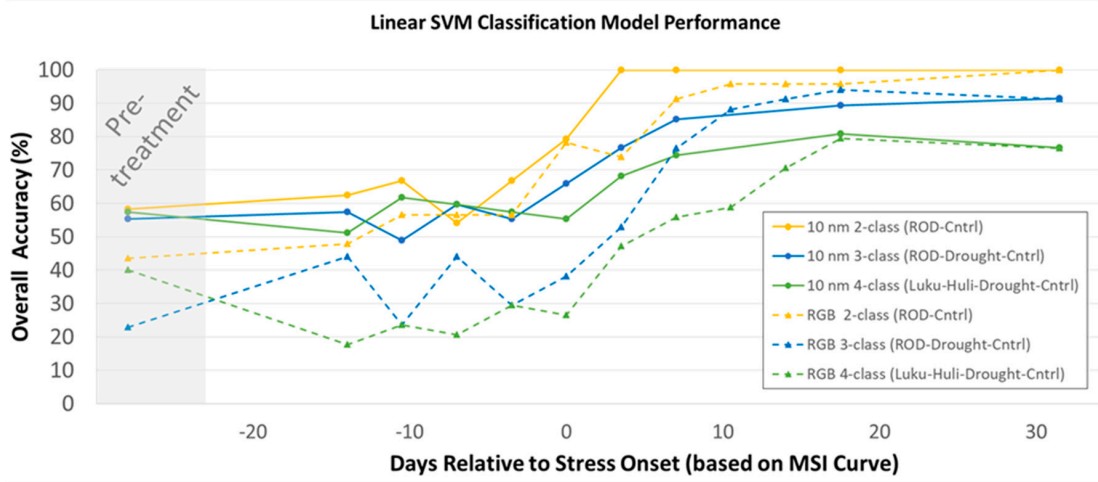

**Figure 13.** Overall support vector machine (SVM) model classification accuracy results for resampled 10 nm-resolution spectral data (ASD 10 nm, solid lines) and visible photographs (RGB, dashed lines) for two-, three-, and four-class models. Models were generated for pre-treatment and different points along the MSI stress curve trajectories, with zero corresponding to stress onset.

Overall accuracies increased with stress stage maturity and decreased with model class size for both datasets. The hyperspectral models largely outperformed the RGB models until the late-stress stage, when model performance eventually converged. At SO, overall accuracies for the hyperspectral two-, three-, and four-class models reached 79.2%, 66%, and 55.3%, with corresponding accuracies of 78.3%, 38.2%, and 26.5% for the RGB models. At one time step prior to SO, accuracies were worse for the two- and three-class models (66.6% and 55.3% for the hyperspectral models, 56.5% and 29.4% for RGB) and essentially unchanged for the poorly performing four-class models. At one time step after SO, under well-established wilting conditions, performance went up significantly for all models (except the two-class RGB, which dropped slightly before rising again in later time steps), reaching 100% accuracy for the two-class hyperspectral model. Increased model performance with symptom stage has been found in other pathogen classification studies [17,20].

Laboratory spectra and RGB imagery of branches collected from the five felled asymptomatic *C. huliohia*-inoculated mature trees were used to test the SO and one time step pre-SO linear SVM models (Table 2). There was generally poor agreement between all models except for tree # 219, where the SO two- and three-class models (and the pre-SO two-class models) all agreed on a ROD classification.

**Table 2.** Linear SVM Model Results for *C. huliohia*-Inoculated Felled Tree Branch Samples.

| Tree | Pre-Stress Onset Model Results | | | | | | Stress Onset Model Results | | | | | |
|---|---|---|---|---|---|---|---|---|---|---|---|---|
| | 2 Class | | 3 Class | | 4 Class | | 2 Class | | 3 Class | | 4 Class | |
| | 10 nm | RGB | 10 nm | RGB | 10 nm | RGB | 10 nm | RGB | 10 nm | RGB | 10 nm | RGB |
| 215 | ROD | Cntrl | Cntrl | D | Cntrl | D | ROD | Cntrl | ROD | Cntrl | D | Cntrl |
| 216 | ROD | Cntrl | D | Cntrl | D | D | ROD | Cntrl | ROD | Cntrl | D | Cntrl |
| 217 | ROD | Cntrl | Cntrl | ROD | Cntrl | luku | Cntrl | ROD | Cntrl | ROD | Cntrl | luku |
| 218 | ROD | Cntrl | ROD | Cntrl | Cntrl | D | ROD | Cntrl | ROD | Cntrl | D | Cntrl |
| 219 | ROD | ROD | D | ROD | Cntrl | luku | ROD | ROD | ROD | ROD | D | Cntrl |

Cntrl = control, D = Drought, luku = C. lukuohia.

## 4. Discussion

### 4.1. Laboratory Trials

The results from this study confirm that the adjective "rapid" is well-placed to describe the expression of these introduced fungal pathogens in infected 'ōhi'a trees. Laboratory seedling trials showed that inoculated 'ōhi'a seedlings quickly experience complete and total wilting following SO, with the more virulent *C. lukuohia* generally being expressed before *C. huliohia*. Time series data collected every 3–4 days from these trials show no immediately obvious pre-SO spectral indicators that could be used for early detection with seedlings, but, instead, two major spectral groupings: pre-SO and post-SO. Wavelengths <725 nm were more sensitive to post-SO changes than wavelengths > 725 nm, suggesting that RGB imagery may be nearly as effective as hyperspectral data for detecting symptomatic ROD trees following SO. Top-down RGB imagery and derived VIs show variation in SO at the leaf scale, with leaves further from the center of the stem generally impacted first (Figure 6). This pattern of fleshy, newly formed stems and leaves on the plants' periphery wilting first is also seen in laurel wilt disease (M. Hughes, personal communication) and is likely due to a lack of lignification and secondary xylem development that provides structural support to these stems [64] and less cuticular waxes that protect leaves from non-stomatal water loss [65]. However, this variation is short-lived and within days all leaves are almost uniformly affected.

VI curves for treated seedlings were generally stable prior to reaching a sharp SO curvature change (Figures 7 and 8), suggesting that VIs are reliable indicators for detecting non-specific stress in 'ōhi'a trees but do not provide significant advance warning of infection in asymptomatic seedlings. There was little difference in the overall shape of the VI curves for the inoculation and extreme drought treatments, except for a somewhat tighter grouping of SO date in the drought treatment seedlings (Figure 7), indicating that VI curves alone cannot be used to discriminate between ROD and extreme drought in a controlled laboratory setting. Outside the laboratory and across the Hawaiian Islands, where extreme drought is atypical and would be well-documented through meteorological measurements, VI curves could be used to identify suspect ROD trees at the SO stage. SO detection via the VI curvature change method proved to be more sensitive than simple thresholding, but requires a dense time series dataset.

Of the six VIs included in this study, MSI and CAI performed marginally better in terms of the early detection of SO (Figure 10), likely due to their component wavelengths being strongly influenced by moisture levels and a cellulose absorption feature near 2100 nm, features also previously identified as important for discriminating healthy and late-stress stage 'ōhi'a leaves [16]. PRI and the RGB-derived VCI were the least stable VIs.

While this study focused on the utility of using available RGB and multispectral imagery for large-scale monitoring programs, an ensemble approach that also integrates information outside of the visible near-infrared (VNIR) portion of the electromagnetic spectrum [22–26] and machine learning-based classification techniques (discussed in Section 4.3) might allow for a more sophisticated early detection of plant stress. In addition, a limitation of our laboratory results was the single independent physical measure of plant health status (a visual rating of seedling wilt condition), which was affected by changes to leaves outside the spectroradiometer measurement area. Targeted sub-canopy measurements of wilt status and other objective measures of plant health, including leaf gas exchange and water status, would be worth including in future work.

### 4.2. Field Trials

The results from the field inoculation trial and sUAS image collection campaign largely mirrored those of the laboratory experiments. Inoculated trees that became symptomatic during the monitoring period degenerated rapidly following SO, with no discernable pre-SO deviations in their VI curves (Figure 12). Within-canopy variation in SO was also present, as shown in Figure 11, with leaves southeast of the central sampling area showing impacts well before the center of the canopy. This suggests that

within-canopy variation in SO could be used to further improve early detection results, given imagery with a fine enough spatial resolution to detect this variation.

The sudden onset of symptoms in ROD-infected trees, particularly for the more virulent *C. lukuohia*, is different from that of other pathogens, where host tree response and/or slow disease progression can compartmentalize and prolong the effects [66–68]. Similar to other vascular wilt diseases, *C. lukuohia* colonizes xylem vessel elements and quickly disperses throughout the host by the movement of spores (conidia) in the sap stream [9,69]. To limit this internal spread, the host may produce tyloses, gels and gums to occlude infected vessels [70,71]. However, this process may result in a drought-like stress and wilt if too many vessels are occluded and no new vessels are formed [29,71]. The abrupt onset of spectral changes in ROD-infected trees may speak to the rapidity of host colonization and the fact that this is not a leaf disease but instead an ailment affecting the vascular system, spectrally hidden until reaching a physiological "tipping point" when wilt and resultant changes in leaf reflectance occur.

Unfortunately, the airborne sensors used in the field component of this study do not have the spectral range or resolution of the full VNIR spectroradiometer used in the laboratory trials. As such, we were unable to collect the same spectrally rich time series dataset in a natural setting. While there was no significant difference in VI performance in the laboratory trials, it is unclear if that would be the case for mature 'ōhi'a trees. The maximum 18 day differential between NDVI and ExG-ExR SO detection for tree # 224 was greater than any of the differences found in the laboratory experiment (maximum of 6 days), and it is possible that similar or greater improvements over NDVI in mature trees could be made by other VIs, including MSI or CAI, the two most promising from the laboratory trials.

In the field trials, the average difference between SO date for the NDVI and ExG-ExR VIs was 4.5 days. Unless that difference can be reliably increased to months instead of days through the use of more sensitive VIs or more powerful classification techniques, it may be difficult to justify the increased complexity of collecting and processing repeat multi- or hyperspectral remote sensing imagery over alternative RGB datasets for a large-scale early detection ROD monitoring program. Other approaches for early detection, including VNIR fusion with thermography and solar-induced fluorescence (SIF) [24,55,56], show great promise for other plant pathogens and should be examined.

Our field results were somewhat complicated by the *C. lukuohia* natural co-infection that occurred in 64% of our *C. huliohia* inoculated trees, causing one of them to become symptomatic immediately following inoculation (tree 211). An additional issue was the tree-felling schedule of the underlying kiln treatment trial, which caused some of the trees to be felled prior to becoming symptomatic. That said, the high number of *C. lukuohia* co-infections further highlight the need to better understand and characterize the progression of this virulent pathogen, and the early tree felling allowed us to collect and analyze leaves from mature pre-symptomatic trees.

### 4.3. Classification Modeling

Our classification efforts, using linear SVM models of various class sizes at different laboratory seedling treatment stages, showed good promise for post-SO stress condition detection (reaching 100%, 91.5%, and 76.6% overall accuracies for the hyperspectral two-, three-, and four-class late-stage models) but were less encouraging for reliable early detection in asymptomatic trees. Model classification accuracies showed few meaningful positive increases from the pre-treatment stage up until the last pre-SO time step, and most model accuracies only markedly improved one time step beyond SO. While these results are from a small number of seedling measurements in a controlled laboratory setting, and therefore do not directly translate into mature trees in the natural environment, they suggest that early detection of ROD infection from remote sensing remains a challenge.

When we applied our classification models to data collected from felled asymptomatic inoculated trees, both two-class hyperspectral models (SO and pre-SO) and the three-class hyperspectral SO model identified all five samples as belonging to the ROD class with one exception, tree 217. These results seem promising, but it is important to note that the leaves from the collected branch may have been quite a bit older than those in the laboratory trials [72] and contained leaf galls and other evidence

of environmental stressors that were not present in the laboratory. In a two-class model, with ROD and the control as the only options, any deviations from young, tender leaves that have been kept in a protected growth chamber at field capacity will be classified as ROD. A more comprehensive data collection and modeling effort, encompassing a larger number of older plants and additional stressors, including persistent drought and ʻōhiʻa rust (*Austropuccinia psidii*) [73,74], is needed to more fully examine the utility of optical and visible near-infrared remote sensing for the identification and early detection of rapid ʻōhiʻa death. Similarly, an exploration of modeling techniques applied to other forest pathogens, including random forest [18] and Extreme Gradient Boosting [75] for myrtle rust (*Austropuccinia psidii*) and partial least-squares discriminant analysis for oak wilt [17], would be worth undertaking in future studies.

## 5. Conclusions

In this study, we characterized spectral progression changes for two ROD fungal pathogens, *C. lukuohia* and *C. huliohia,* through laboratory seedling treatment trials, frequent spectroradiometer measurements and RGB photography. We compared these results with data collected under extreme drought conditions and found that the ROD symptoms appeared very abruptly, with little obvious spectral development prior to the visible onset of stress. A number of different VIs were calculated from the datasets; of these, MSI and CAI were found to be marginally more sensitive in detecting the early onset of stress, though never more than five days in advance of visual detection. Similar results were found for the sUAS-derived field trial data, though we only compared two VIs (NDVI and ExG-ExR). Alternative VIs, including ʻōhiʻa-optimized versions of MSI and CAI (Supplemental Figure S4), may extend the early detection window beyond what was achieved with NDVI, but it seems unlikely that gains greater than six months may be achieved with VIs over visible indicators, given the abrupt onset of ROD symptoms and sensitivity in wavelengths <725 nm.

We developed a sequence of linear SVM classification models from the laboratory trial data and found that model performance improved with advancing stress stage, as has been found in similar studies, but also that these models had difficulties reliably identifying asymptomatic trees, including when applied to data collected from mature field trees with leaf conditions absent in the modeling database. We believe improved results may be obtained with additional datasets and the exploration of other types of classification models, but the abrupt onset of ROD symptoms will remain a fundamental challenge for early detection.

**Supplementary Materials:** The following are available online at http://www.mdpi.com/2072-4292/12/11/1846/s1, Figure S1: VI curves for inoculation seedlings (colors and line definitions are same as Figure 8), Figure S2: VI curves for drought seedlings (colors and line definitions are same as Figure 8), Figure S3. VI curves (NDVI and MSI) from incremental curve fitting data analysis to determine the impact of partial data on the successful detection of stress onset (SO) for four seedlings, Figure S4. Spectral Separability Index (SSI) evolution over time for different treatments and increasing stress condition, Table S1: Linear SVM classification model results.

**Author Contributions:** Conceptualization, R.L.P., M.H. and L.M.K.; Data curation, R.L.P.; Formal analysis, R.L.P.; Funding acquisition, R.L.P.; Investigation, R.L.P., M.H., E.C., T.S. and G.L.; Methodology, R.L.P., M.H., L.M.K., E.C., T.S. and G.L.; Project administration, R.L.P.; Resources, R.L.P. and L.M.K.; Software, R.L.P.; Supervision, R.L.P.; Validation, R.P.; Visualization, R.P. and E.C.; Writing—Original draft, R.P.; Writing—Review & editing, R.L.P., M.H., L.M.K., E.C. and T.S. All authors have read and agreed to the published version of the manuscript.

**Funding:** This research was funded by Conservation X Labs and the Hawaii Department of Land and Natural Resources, and used instrumentation purchased via grants from the National Science Foundation, No. (1839095 and 1828799).

**Acknowledgments:** We thank the ROD Science Group and Big Island Invasive Species Committee for sample collection and testing. We thank Wade Heller, Eva Brill, and Lionel Sugiyama for their invaluable technical assistance. We thank Sharon Dansereau, William Emery, Keely Roth, and Carl Legleiter for reviewing the manuscript and providing valuable suggestions. Mention of trademark, proprietary product, or vendor does not constitute a guarantee or warranty of the product by the U.S. Dept. of Agriculture and does not imply its approval to the exclusion of other products or vendors that also may be suitable.

**Conflicts of Interest:** The authors declare no conflict of interest. The funders had no role in the design of the study; in the collection, analyses, or interpretation of data; in the writing of the manuscript, or in the decision to publish the results.

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
