# Peer review of "Examining the Utility of Visible Near-Infrared and Optical Remote Sensing for the Early Detection of Rapid ‘Ōhi‘a Death"

_remotesensing, doi:10.3390/rs12111846_

Round 1
Reviewer 1 Report
This study tries to remotely detect the effects of two plant pathogens using lab and field measurements. The manuscript is well written, and the results are clearly presented. The conclusion is that the early detection of the pathogen effects using remote sensing is still challenge, especially at landscape scales. Personally, I tend to agree with the authors, but a few newly developed methods might have to be tested or at least discussed to get a more convincible result.
Conventional vegetation indices vs newly developed metrics
Although a field spectrometer was used, this work mainly used a few vegetation indices including NDVI and PRI. I like the idea of starting from the simple indices, but it would be better to test some other methods (for example, some metrics in Dr. Pablo Zarco-Tejada’s work as cited in this manuscript) since they have been reported as ‘useful.’
Plant health assessment
This work relied on the visual estimation of the plant health status. I think it would be better to have some other objective measurements (such as photosynthesis rate using gas exchange or fluorescence) to indicate the actual plant status particularly for the experiment that was done under lab environments. I understand that it is hard to redo everything, but I feel this part is worth being discussed at least.
Within canopy variation
The reflectance and camera only covered part of the plants. During the visual inspection, foliar wilt that was outside the FOV of spectrometer was also counted. Another thing is the canopy reflectance is sort of averaged spectrum. I am curious about the time series spectral patterns of different leaves rather than the whole canopy since this may change the timing of the detection.
A lot of the ‘Discussion’ reads more like ‘Results’
In the introduction, the authors talked about some successful remote sensing applications in the pathogen detections. It might be good to come back to this topic later in the manuscript to discuss what methods might /might not work and why in more details. Meanwhile, discussions on multispectral (red-NIR) vs hyperspectral, thermal & SIF might be needed.
Minor comments
Line 51: It might be better to label trees in different infected stages directly in the photo.
Line 82: Fig 2
Line 282: I may have missed something, but I am not sure if I understand the figure legend. For example, for panel A, what is the difference between ‘Day 16’ and ‘Day-16?’
Line 318: Sorry, I cannot find the supplement materials for some reason.
Line 325: Could you please consider using some sort of boxplot to show the variation in the timing of the stress onset?
Line 341: I wonder if it is due to the relatively small sample size.
Author Response
Dear Reviewer #1, we thank you for your comments and suggestions, please see our responses below and in the revised submitted document.
Conventional vegetation indices vs newly developed metrics
Although a field spectrometer was used, this work mainly used a few vegetation indices including NDVI and PRI. I like the idea of starting from the simple indices, but it would be better to test some other methods (for example, some metrics in Dr. Pablo Zarco-Tejada’s work as cited in this manuscript) since they have been reported as ‘useful.’
Unfortunately, we did not collect the physical measurement data necessary to use the “hybrid wavelet-inverted model inversion method” Zarco-Tejada et al. uses in 2018 paper. We did attempt to calculate and include the Normalized Phaeophytinization Index (????), as this index was new to us and found by Zarco-Tejada et al. to be the most powerful VI for early detection in their study, but the short wavelengths involved in the index (?415 and ?435) are not well measured by our spectroradiometer instrument (measured data <450 nm are quite noisy). We do recognize that a more sophisticated approach could be used and have added additional text in the Discussion along these lines (lines 519-523)
Plant health assessment
This work relied on the visual estimation of the plant health status. I think it would be better to have some other objective measurements (such as photosynthesis rate using gas exchange or fluorescence) to indicate the actual plant status particularly for the experiment that was done under lab environments. I understand that it is hard to redo everything, but I feel this part is worth being discussed at least.
We agree that it would be very useful to have additional physical measurements of plant health status beyond the visual estimation of wilting, but do not think that they are required for publication. The central aim of this study was to examine the potential for using hyperspectral and RGB (red-green-blue) visible-wavelength imagery at the leaf and branch scale to detect ROD prior to the onset of visible symptoms. Asner et al. 2018 have already established strong connections between leaf chemistry and spectral signatures in healthy and late-stage ROD-infected ‘ōhi‘a trees, including changes in chlorophyll, water, and non-structural carbohydrates, and we did not think it was necessary to repeat that work, though in retrospect additional physical measurements would have allowed us to delve deeper into the mechanisms for stress onset and why they appear so suddenly in ROD-infected trees.
Additional text added, lines 523-527
Within canopy variation
The reflectance and camera only covered part of the plants. During the visual inspection, foliar wilt that was outside the FOV of spectrometer was also counted. Another thing is the canopy reflectance is sort of averaged spectrum. I am curious about the time series spectral patterns of different leaves rather than the whole canopy since this may change the timing of the detection.
The spectroradiometer data did indeed only capture part of the seedling canopy, but top-down laboratory RGB camera images capture the whole seedlings, not just the portion measured by the spectroradiometer.
Figure 6 has been revised to include ExG-ExR images, which are derived from the native RGB images and highlight changes at the individual leaf scale across the entire seedling canopy. Days 37, 41, and 44 in particular show variation across different leaves, with leaves further from the center of the stem generally impacted first. This pattern is similar to that seen for the sUAS data shown in Figure 11, where the NDVI spatial pattern at days 63 and 73 shows lower NDVI values immediately southeast of the central sampling area. With fine enough spatial resolution imagery, this within-canopy variation can be used to improve early detection results.
Additional explanatory text has been added to the Discussion section, lines 500-502
A lot of the ‘Discussion’ reads more like ‘Results’
Additional text has been added throughout the Discussion to help the tone.
In the introduction, the authors talked about some successful remote sensing applications in the pathogen detections. It might be good to come back to this topic later in the manuscript to discuss what methods might /might not work and why in more details. Meanwhile, discussions on multispectral (red-NIR) vs hyperspectral, thermal & SIF might be needed.
Additional text has been added, lines 567-568, though we are not sure at the present time which of these methods might or might not work
Minor comments
Line 51: It might be better to label trees in different infected stages directly in the photo.
We replaced the photo in Figure 1 with one that more clearly showed trees in different stages of infection and labeled examples. Additional information about the location and date of the photo was also added to the caption.
Line 82: Fig 2
??
Line 282: I may have missed something, but I am not sure if I understand the figure legend. For example, for panel A, what is the difference between ‘Day 16’ and ‘Day-16?’
Added additional text in the caption explaining legend:
“Numbers in legend refer to days since treatment began.”
Line 318: Sorry, I cannot find the supplement materials for some reason.
We originally submitted these in a separate file as instructed during the upload process, have now added them back in at the end of the submitted revised document.
Line 325: Could you please consider using some sort of boxplot to show the variation in the timing of the stress onset?
Figure 10 contains boxplots showing the variation in timing of stress onset by treatment and has been re-labeled to increase clarity. In re-examining this figure we discovered that the drought stress onset data used to make the original figure erroneously contained rows of zeroes. These have been removed, resulting in a different panel B.
Additional explanatory text interpreting Figure 10 has been added in lines 380-383
Line 341: I wonder if it is due to the relatively small sample size.
Perhaps, but we think it is more likely because the seedlings are small in stature and the effects of the treatments hit them quickly
Reviewer 2 Report
Comments to the authors of the paper: Examining the utility of visible near-infrared and optical remote sensing for the early detection of rapid ōhi‘a death
General comments:
This paper presented investigation into the potentials of using frequent laboratory and field measurements to detect pre-symptomatic trees infected with these pathogens in the visible near-infrared and optical remote sensing. Generally, this manuscript is well written and the topic of the paper is interesting for potential readers. Thus, it is probably appropriate for this journal. I would suggest accepting this paper for publication after major revisions. Since I don’t have enough time to closely check everything for this manuscript, here I mainly focus on some of my major concerns and comments:
- This paper made efforts on the work from laboratory and field inoculation trials to detect so-called ROD prior to the onset of visible symptoms using hyperspectral and visible imagery at the leaf and branch scale, which is a merit for this paper. However, it seems to me that new methodology has been not proposed and developed in this manuscript. Indeed, the authors should unambiguously point out the novelty of their methodology in their study. This is one of my major concerns for this study.
- In general, it seems to me that this manuscript is not easy to follow up. I suggest the authors provide a general flow chart to logically illustrate the process of their study, which will most probably facilitate readers to follow and understand their study.
- I think this manuscript focuses on much more experimental details and processes, mainly providing figures to describe their experiment results; however, the analysis of their results is insufficient. I suggest the authors strengthen the analysis for their results. Although this paper provided few analyses in the Discussion part, these analyses should be extended and enforced by directly following the descriptions of their result for each figure and table.
- I suggest that authors provide more details about their classification model, method and process. The current classification part, i.e., section 3.3. is not clear to me. In addition, I failed in acquiring the supplemental figures and table from the given website. I suggest these supplemental figures and table be provided directly at the end of manuscript.
Minor comments
- The figures and table in this manuscript seem okay to me. But Figure 1 should provide more details, e.g., geographical latitude/longitude, altitude, acquisition time etc.. A similar requirement is for Figure 4.
- The manuscript should be carefully checked by authors. Although I did not spare enough time to check everything, but there are still some possible typos that need to be confirmed, e.g., Line 154: “caretonoid” should be carotenoid? the legend of Figure 7, “lukuoia” should be “lukuohia”? Line 341: what is ANOVA? etc..
Author Response
Dear Reviewer #2, we thank you for your comments and suggestions, please see our responses below and in the revised submitted document.
- This paper made efforts on the work from laboratory and field inoculation trials to detect so-called ROD prior to the onset of visible symptoms using hyperspectral and visible imagery at the leaf and branch scale, which is a merit for this paper. However, it seems to me that new methodology has been not proposed and developed in this manuscript. Indeed, the authors should unambiguously point out the novelty of their methodology in their study. This is one of my major concerns for this study.
There are at least two areas where this manuscript is novel. The first is the attempt to detect infection of the fungal pathogens responsible for ROD in asymptomatic trees using VNIR remote sensing, which has not been done before for this vascular pathogen.
This is mentioned in lines 62-64
A second point of novelty is a new approach for detecting stress onset using partial time-series datasets, something that was suggested by another reviewer and is now included in the manuscript. This is introduced in lines 190-194
We believe that these two points together provide enough novelty for this paper to be published.
- In general, it seems to me that this manuscript is not easy to follow up. I suggest the authors provide a general flow chart to logically illustrate the process of their study, which will most probably facilitate readers to follow and understand their study.
We appreciate this comment but do not think that the addition of another figure in the main text is required. Instead, we added and revised text throughout the paper to make it easier to follow. These additions can be seen in the track changes version of the submitted manuscript.
- I think this manuscript focuses on much more experimental details and processes, mainly providing figures to describe their experiment results; however, the analysis of their results is insufficient. I suggest the authors strengthen the analysis for their results. Although this paper provided few analyses in the Discussion part, these analyses should be extended and enforced by directly following the descriptions of their result for each figure and table.
We have strengthened the Results and Discussion sections by modifying figures to make them more informative and by adding additional explanatory text, particularly lines 380-392, and throughout the Discussion section.
- I suggest that authors provide more details about their classification model, method and process. The current classification part, i.e., section 3.3. is not clear to me.
We appreciate the reviewer’s comments on this point and have made changes and additions to sections 2.7 and 3.3 to better explain the classification models, method, and process.
Rather than train the models using stressed and unstressed samples, we were interested to see how well our models could discriminate between classes at different stress stages, so we created different models for the different stress stages, using the seedling MSI curves as the basis for selecting the appropriate input data for the different models. We hypothesized that we would be unable to discriminate between treated and untreated seedlings at the pre-treatment stage, since at that point they should all be the same (healthy). Similarly, we hypothesized that we would have little trouble discriminating between inoculated and control trees at late stress stages, for example, because the control trees would remain healthy and green while the inoculated trees would be experiencing late-stage wilting. These hypotheses are largely borne out in Figure 13, though the >50% accuracy for different 10 nm spectroradiometer models at the pre-treatment stage was unexpected and suggests that there were some slight differences among the seedling groups from the very start that the spectroradiometer was able to detect.
We saw little improvement in classification model accuracy from the pre-treatment stage up to the time step just before SO, which was disappointing. At SO and afterwards, we did see significant increases in model accuracies, but from an early-detection standpoint making correct classifications at these stages is unfortunately too little too late.
Additional text better explaining our approach has been added to sections 2.7 Treatment Discrimination via Classification and 3.3 Classification Model results
In addition, I failed in acquiring the supplemental figures and table from the given website. I suggest these supplemental figures and table be provided directly at the end of manuscript.
We originally submitted these in a separate file as instructed during the upload process, have now added them back in at the end of the submitted revised document.
Minor comments
- The figures and table in this manuscript seem okay to me. But Figure 1 should provide more details, e.g., geographical latitude/longitude, altitude, acquisition time etc.. A similar requirement is for Figure 4.
This information has been provided in revised versions of Figures 1 and 4
- The manuscript should be carefully checked by authors. Although I did not spare enough time to check everything, but there are still some possible typos that need to be confirmed, e.g., Line 154: “caretonoid” should be carotenoid? the legend of Figure 7, “lukuoia” should be “lukuohia”? Line 341: what is ANOVA? etc..
These have been fixed, thank you for identifying these errors
Reviewer 3 Report
This study deals with an important topic, it is based on meticulous and well-documented lab- and fieldwork, and it is well organized and well presented in the paper. I am a proponent of presenting somewhat disappointing results like these, they are very important in giving future researchers a point of departure (i.e. no need to repeat things that didn't work well), and they give land managers important information about what is feasible and what isn't.
I made a number of minor comments in the text, but for the most part the writing and presentation are very good. I have two more general comments that follow below. Both could be addressed by either a minor revision (adding some explanatory text) or a moderate revision (re-running some analyses).
1. The machine classification models. I didn't quite understand how these were set up. My approach would be to train the model using stressed and unstressed samples and then see how well the model recognized these classes in a larger data set. I don't think that is how the author's did it here, because if so the results should have shown high accuracy prior to treatment (when all samples are healthy, and the model does well at recognizing that), a dip in accuracy during the subtle early stress time (when the model has trouble detecting the subtle changes), and then high accuracy again at the end when all of the samples are obviously either fine (the controls) or highly stressed. This isn't how the model results look in the paper (Fig. 13) so I guess the model structure was not like this, but am unsure what it was. Some added explanation to the Methods and Results are needed here (i.e. a fairly minor revision to the paper), and the authors might consider re-running the model using the approach that I just outlined (a moderate revision).
2. Early detection vs. retrofitting. The ultimate goal here is recognizing infections early on, before it has progressed very far. The two metrics used here (the abrupt curvature change point prior to the stress center and the 20% threshold in the stress direction) require data to be collected well past stress onset, so that the curve can be fit or the 20% threshold value can be identified. So they require that you have nearly the full time series (and the tree is already essentially dead) before you can find the date of stress onset. In other words, they are in a sense only useful for retrofitting a curve after the fact, not for early detection. Its true that if you can't find an early indicator even by retrofitting with the full curve available, finding an early indicator in real time will be even harder. So in that sense the limitations for early detection are even more severe than what their results indicate. To deal with this issue the authors should at least explain the quandry (i.e. a minor revision to the paper) or simulate "real-time" (i.e. partial) curve fitting and see how it works (i.e. a moderate revision to the paper). My approach to the latter would be to use the full curve-fitting results to choose the most promising method for recognizing the stress onset date, and then attempt to locate the dates using time series that are incrementally lengthened from day 1 through the end of the data set. You might get some false early SO dates (e.g. a transient dip is identified as the real thing) or failure to recognize the SO until you are well past it because you need the long curve to even register the inflection.

Author Response
Dear Reviewer #3, we thank you for your comments and suggestions, please see our responses below and in the revised submitted document.
- The machine classification models. I didn't quite understand how these were set up. My approach would be to train the model using stressed and unstressed samples and then see how well the model recognized these classes in a larger data set. I don't think that is how the author's did it here, because if so the results should have shown high accuracy prior to treatment (when all samples are healthy, and the model does well at recognizing that), a dip in accuracy during the subtle early stress time (when the model has trouble detecting the subtle changes), and then high accuracy again at the end when all of the samples are obviously either fine (the controls) or highly stressed. This isn't how the model results look in the paper (Fig. 13) so I guess the model structure was not like this, but am unsure what it was. Some added explanation to the Methods and Results are needed here (i.e. a fairly minor revision to the paper), and the authors might consider re-running the model using the approach that I just outlined (a moderate revision).
Rather than train the models using stressed and unstressed samples, we were interested to see how well our models could discriminate between classes at different stress stages, so we created different models for the different stress stages, using the seedling MSI curves as the basis for selecting the appropriate input data for the different models. We hypothesized that we would be unable to discriminate between treated and untreated seedlings at the pre-treatment stage, since at that point they should all be the same (healthy). Similarly, we hypothesized that we would have little trouble discriminating between inoculated and control trees at late stress stages, for example, because the control trees would remain healthy and green while the inoculated trees would be experiencing late-stage wilting. These hypotheses are largely borne out in Figure 13, though the >50% accuracy for different 10 nm spectroradiometer models at the pre-treatment stage was unexpected and suggests that there were some slight differences among the seedling groups from the very start that the spectroradiometer was able to detect.
We saw little improvement in classification model accuracy from the pre-treatment stage up to the time step just before SO, which was disappointing. At SO and afterwards, we did see significant increases in model accuracies, but from an early-detection standpoint making correct classifications at these stages is unfortunately too little too late.
Additional text better explaining our approach has been added to sections 2.7 Treatment Discrimination via Classification and 3.3 Classification Model results
- Early detection vs. retrofitting. The ultimate goal here is recognizing infections early on, before it has progressed very far. The two metrics used here (the abrupt curvature change point prior to the stress center and the 20% threshold in the stress direction) require data to be collected well past stress onset, so that the curve can be fit or the 20% threshold value can be identified. So they require that you have nearly the full time series (and the tree is already essentially dead) before you can find the date of stress onset. In other words, they are in a sense only useful for retrofitting a curve after the fact, not for early detection. Its true that if you can't find an early indicator even by retrofitting with the full curve available, finding an early indicator in real time will be even harder. So in that sense the limitations for early detection are even more severe than what their results indicate. To deal with this issue the authors should at least explain the quandry (i.e. a minor revision to the paper) or simulate "real-time" (i.e. partial) curve fitting and see how it works (i.e. a moderate revision to the paper). My approach to the latter would be to use the full curve-fitting results to choose the most promising method for recognizing the stress onset date, and then attempt to locate the dates using time series that are incrementally lengthened from day 1 through the end of the data set. You might get some false early SO dates (e.g. a transient dip is identified as the real thing) or failure to recognize the SO until you are well past it because you need the long curve to even register the inflection.
We think this is a very good point and ran this analysis for two of the VIs (NDVI and MSI) from the seedling trial data to determine how important it is to have a complete vs partial dataset for SO detection. For this analysis we used a slightly different approach from what was originally presented in the paper. Instead of working out from the “center” of the known active stress period, here we simply searched for the first appearance of an abrupt change in signal, beginning with three time-series data points and used an incrementing for loop to increase the amount of data available for analysis, resulting in progressively longer curves to search. Results from seedlings #1-#4 are shown in Supplemental Figure S3 (curves offset for clarity). We found that the detection of SO using this method was quite stable in cases with well-behaved curves, only varying by 1-2 days, and typically within a few days of what was found with the original approach. Thus it is possible to catch the SO signal before the tree is completely dead using this technique, but likely not soon enough to save the tree. In more noisy datasets, as for Seedling #3 (Figure S3C), we did see early false triggers and more spread in SO dates.
Reviewer 4 Report
Summary and contribution
The authors analyze spectral properties of fungus-affected ohia trees through time with laboratory and field tests. Their research was motivated by the need to develop an early detection method for rapid ohia death. While they were able to detect stress in general before ohia tree death, the authors were not able to detect infected, but asymptomatic trees more than 5 days in advance of visual wilting.
Although the authors report negative results, the task of early detection of pathogens in asymptomatic trees is a difficult task, and negative results are still valuable. Negative results were not the result of a poor analysis; the authors’ analysis was thorough and well done. Those interested in the early detection of tree pathogens with remote sensing should find this article interesting and useful.
Comments
I could find no major issues with the manuscript. It is well written and easy to read. The introduction contains sufficient background material, and the research is well motivated. The methods are sound and described well enough that they could be repeated. The combination of the laboratory and field tests makes for a thorough analysis. The authors’ conclusions follow from their results.
Table 1. Is there a reason ‘r’, ‘g’, and ‘b’ are not capitalized in the ExG-ExR formula, but are capitalized for the VCI formula?
Figure 4. I recommend labeling which island is shown on Figure 4.
I was not able to view the authors’ supplementary figures. If they are like the authors’ other figures, they are well made. I’m guessing they are fairly large, so it makes sense to keep them out of the main body of the text.
Figure 12. I’m guessing the journal will want these panels labeled with ‘A’ and ‘B’. Also, on the top panel, it probably goes without saying, but these lines represent mean VI curves, correct?
The authors have 13 figures. I wouldn’t drop any of them, since they are all useful. However, Figures 6 and 8 are less useful in my opinion and could be dropped to make the article shorter.
Author Response
Dear Reviewer #4, we thank you for your comments and suggestions, please see our responses below and in the revised submitted document.
Table 1. Is there a reason ‘r’, ‘g’, and ‘b’ are not capitalized in the ExG-ExR formula, but are capitalized for the VCI formula?
These letters have now been made uniform in capitalization
Figure 4. I recommend labeling which island is shown on Figure 4.
This label has been added and the Figure edited to include lat/lon information
I was not able to view the authors’ supplementary figures. If they are like the authors’ other figures, they are well made. I’m guessing they are fairly large, so it makes sense to keep them out of the main body of the text.
We originally submitted these in a separate file as instructed during the upload process, have now added them back in at the end of the submitted document.
Figure 12. I’m guessing the journal will want these panels labeled with ‘A’ and ‘B’. Also, on the top panel, it probably goes without saying, but these lines represent mean VI curves, correct?
Labels have been added to this figure. These lines represent NDVI and ExG-ExR curves for individual trees as measured from sUAS imagery collected over our field sites, and are the averaged values from pixels included in a 1 m diameter region of interest over the center of the tree canopy.
The authors have 13 figures. I wouldn’t drop any of them, since they are all useful. However, Figures 6 and 8 are less useful in my opinion and could be dropped to make the article shorter.
We thank the reviewer for this suggestion but have decided to keep both these figures, including an edited version of Figure 6 based on another reviewer’s comments. We did add one more Supplemental figure.
Round 2
Reviewer 1 Report
Thanks for providing the responses. I'm pleased with the current form of the manuscript.
Congratulations on the work!
Author Response
Thank you
Reviewer 2 Report
Thanks the authors for considering my comments. Because of the direct addition of necessary figures and tables in Appendix, a flowchart is not necessarily required in current version. I think the current manuscript has been improved; thus I suggest accept this paper for publication in current stage.
Author Response
Thank you